**Technical Note: Monitoring discharge of mountain streams by**
**retrieving image features with deep learning**
Chenqi Fang[1], Genyu Yuan[1], Ziying Zheng[1], Qirui Zhong[1], Kai Duan[1*]
*[1] School of Civil Engineering, Sun Yat-Sen University, Guangzhou, China*
*Corresponding author, E-mail: duank6@mail.sysu.edu.cn

# Abstract

Traditional discharge monitoring usually relies on measuring flow velocity and cross-section area with various velocimeters or remote-sensing approaches. However, the topography of mountain streams in remote sites largely hinders the applicability of velocity-area methods. We here present a method to continuously monitor mountain stream discharge using a low-cost commercial camera and deep learning algorithm. A procedure of automated image categorization and discharge classification was developed to extract information on flow patterns and volumes from high-frequency red–green–blue (RGB) images with deep convolutional neural networks (CNNs). The method was tested at a small, steep, natural stream reach in southern China. Reference discharge data was acquired from a V-shaped weir and ultrasonic flowmeter installed a few meters downstream of the camera system. Results show that the discharge-relevant stream features implicitly embedded in RGB information can be effectively recognized and retrieved by CNN to achieve satisfactory performance in discharge measurement. Coupling CNN and traditional machine learning models (e.g., support vector machine and random forest) can potentially synthesize individual models' diverse merits and improve generalization performance. Besides, proper image pre-processing and categorization are critical for enhancing the robustness and applicability of the method under environmental disturbances (e.g., weather and vegetation on river banks). Our study highlights the usefulness of deep learning in analyzing complex flow images and tracking flow changes over time, which provides a reliable and flexible alternative

apparatus for continuous discharge monitoring of rocky mountain streams.
**Keywords**:
Discharge monitoring; Mountain streams; Deep learning; Machine learning; Image
categorization

# 31 1 Introduction

Continuous discharge data is critical for hydrological model development and flood
forecast (Clarke, 1999; Mcmillan et al., 2010), water resources management (Council,
2004), and aquatic ecosystem health assessment (Carlisle et al., 2017). Traditional
discharge monitoring relies on stream gauges that convert water level to discharge with
an established stage-discharge curve, or information on stable cross-sections and flow
velocity obtained from flow velocimeters such as Acoustic Doppler Current Profiler
(ADCP) and ultrasonic defectoscope (Kasuga et al., 2003). However, these approaches
require significant investment on the implementation of equipments, training of
personnel with expertise, and constant maintenance (Fujita et al., 2007; Czuba et al.,
2017; Yorke and Oberg, 2002). Besides, the performance of transducers and
velocimeters is usually susceptible to sediments and floating debris, particularly in
flooding seasons (Hannah et al., 2011). Consequently, large temporal gaps remain in
many discharge records across the world despite of the growing demand on data
(Davids et al., 2019; Royem et al., 2012). Spatially, flow monitoring of downstream
river sections has been assigned to a higher priority due to the concerns on water supply
and flood control, leading to an acute shortage of discharge data in mountain streams
and headwater catchments (Deweber et al., 2014).

To overcome the limitations of traditional methods, a few image-based approaches

have been introduced into water stage, flow velocity, and discharge measurement in
rivers (Noto et al., 2022; Leduc et al., 2018). Image-based (Leduc et al., 2018; Noto et
al., 2022) approaches rely only on the acquisition of digital images of streams from
inexpensive commercial cameras and thus have been a promising alternative for
continuous, noninvasive, and low-cost streamflow monitoring. The two most
commonly used approaches include large-scale particle image velocimetry (LSPIV)
and particle tracking velocimetry (PTV). LSPIV (Fujita et al., 2010) is based on a high-
speed cross-correlation scheme between an interrogation area (IA) in a first image and
IAs within a search region (SR) in a second image. The technique has been proved
effective in monitoring low-velocity and shallow-depth flow fields (Tauro et al., 2018).
However, it performs poorly in mapping velocity fields in high resolution when there
is a lack of seeds on the water surface because the algorithm obtains the average speed
of each SR (Tauro et al., 2017). Compared to LSPIV, PTV was designed for low seeding
density flows, focusing on particle tracking instead of recognition. The PTV approach
does not require assumptions on flow steadiness nor the relative position of neighbor
particles (Tauro et al., 2018). Several algorithms have been developed for PTV analysis,
such as space-time image velocimetry (STIV) and optical tracking velocimetry (OTV),
overcoming the over-dependence on natural particles' shape and size (Tauro et al., 2018;
Tsubaki, 2017). STIV evaluates surface flow velocity by analyzing a texture angle
within a variation of brightness or color on the water surface, while OTV combines
automatic feature detection, Lucas-Kanade tracking algorithm and track-based filtering
methods to estimate subpixel displacements (Fujita et al., 2007; Karvonen, 2016).
Existing image-based discharge measurement methods all use the velocity-area method
to indirectly deduce discharge after identifying stage and average (Davids et al., 2019;
Leduc et al., 2018; Tsubaki, 2017; Herzog et al., 2022) velocity. The average velocity
in a cross-section is estimated with surface velocity derived from natural or artificial
seeds on water surface and pre-defined empirical relationships between the surface
velocity and average velocity. The velocity-area method relies on a stable relationship
between stage and cross-sectional area, and needs to take velocity extrapolations to the
edges and vertical distributions throughout the cross-section into account (Le Coz et al.,
2012). However, it is difficult to identify the water stage and vertical characteristics of
mountain streams due to the steep, narrow, and highly heterogeneous cross-sections.
The applicability of PIV and PTV approaches is largely hindered by such topography.
Unlike PIV and PTV, deep learning models possess the capability to extract
discharge-related features from images of rivers or streams automatically. These models
are able to adjust the weights assigned to each feature, eliminating the need for manual
attention and reducing the risk of overemphasizing or misinterpreting features that are
unresponsive to flow discharge (Canziani et al., 2016). Besides, deep learning models
can extract low-level image features, such as edges, textures, and colors (Jiang et al.,
2021). These merits could be essential in retrieving information from images of
mountain streams, particularly in regions with intricate cross-sectional profiles. For
example, Ansari et al. (2023) developed a convolutional neural network (CNN) to
estimate the spatial surface velocity distribution and derive discharge, outperforming
traditional optical flow methods both in laboratory and field settings, albeit with a
reliance on surveyed cross-section information.
In this study, we propose a novel mountain stream discharge monitoring method
using a low-cost commercial camera and deep learning models. Automated image
categorization and pre-processing procedures were developed for processing high-
frequency red–green–blue (RGB) images, and then CNN was used to extract
information on flow patterns from RGB matrices and establish empirical relationships
with the classification probabilities of discharge volumes. We hypothesize that (1) the
features of mountain streams (e.g., coverage of water surface, flow direction, flow
velocity) embedded in RGB images can be recognized by suitable deep learning
approaches to achieve effective discharge monitoring, and (2) proper image pre-
processing and categorization can improve accuracy of image-based discharge
monitoring of mountain streams. A rocky mountain stream of a headwater catchment in
tropical southern China was used as a study site to test our hypotheses.

## 108  2 Methods

### 109  2.1 Site and field setting

The study site is located on a small, steep, rocky reach of a stream in the Zhuhai Campus
of Sun Yat-sen University, China (22°20′58″ N, 113°34′29″ E). The site elevation is 13
m above sea level and about 2 km away from the Lingding Yang of South China Sea.
The stream flow is mainly controlled by rainfall in the upstream drainage area. Water
stage and flow velocity increase rapidly during East Asian summer monsoon rainfalls
and fluctuate with synoptic weather conditions on dry days.

The main objective of the study was to test the applicability of deep-learning based

image processing approaches in capturing the flow characteristics and discharge
volumes in the daily flow cycle in this mountain stream. We selected a straight, single-
thread reach for the gauging location, and set up a Hikvision camera on the left bank of
the stream to collect flow images (**Fig. 1**). Discharge data monitored by a weir about 8
m downstream of the camera was used for model training and validation. The camera
was installed 3 m above the ground, facing the surface of the stream almost vertically.
The entire stream width is visible in the images. The camera was equipped with a 150W
solar panel and 80AH lithium battery, enabling the camera to work continuously for 80
hours without external power on rainy days. The camera supports the wireless
transmission of video data to the server.

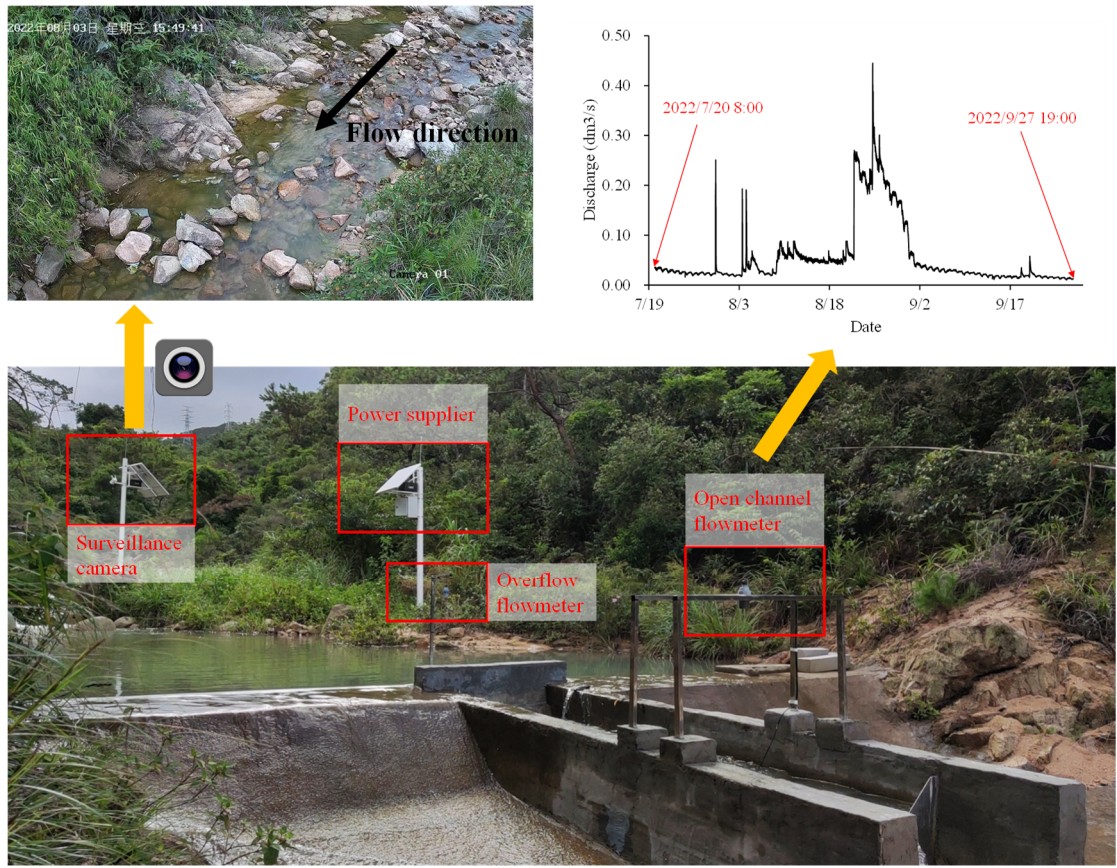


**Figure 1.** Camera setup. The camera is set on the left bank of the stream, about 3 m

above the water surface, and 8 m upstream of a gauging weir. The top right panel

demonstrates the changes in the flowmeter's discharge during the measurement period.


**2.2 Data**

The flat V-shaped weir downstream of the camera monitors discharge with an open

channel flowmeter and an overflow flowmeter. The flowmeters measure water levels in

the channel and in front of the weir with ultrasonic sensors and calculate real-time

discharge at the time step of two minutes by a semi-empirical equation suggested by

the State Bureau of Technical Supervision of China (www.chinesestandard.net), as

$$\qquad Q = \frac{8}{15} C_e \tan\frac{\theta}{2} \sqrt{2g} h_e^{\frac{5}{2}} \qquad\qquad (1)$$
where $Q$ is the discharge of stream, $\theta$ is the angle of triangular weir, $g$ is
acceleration of gravity, $h_e$ is the height of the water surface from the bottom of triangle
barrier, $C_e$ is an empirical coefficient.
We collected the discharge data of the weir (**Fig. 1**) and its corresponding stream
videos during daylight (07:00-19:00 UTC+8) from July 20th to September 27th, 2022.
The raw video resolution was 2560×1440 pixels with a refresh rate of 50 Hz. Images
were extracted from the videos at 5-minute intervals to avoid excessive similarity
between adjacent images. A total of 7,757 image samples labeled with 37 discharge
values between 0.014 and 0.050 m³/s at the interval of 0.001 m³/s were collected for
model testing.
**2.3 Image processing**
**2.3.1 Image categorization**
Environmental disturbances such as illumination and shadow can seriously interfere
with the extraction of effective image features of mountain streams, such as boundaries
of water surface and textures of flow lines (Herzog et al., 2022; Gershon et al., 1986).
Although researchers have proposed methods to eliminate shadows (Finlayson et al.,
2002), the treatment effect in some complex environments, such as plant shadows and
boulders distributed on mountain streams, is not always satisfactory.
Frequently observed disturbances on images include: (1) shadows in the target stream
region due to plants blocking direct sunlight; (2) image noise due to raindrops attached
to the camera lens on rainy days; (3) the lack of light leading to low brightness and
contrast of the image; (4) overexposure of image due to light reflection of the water
surface (around 16:00 UTC+8 in this case). Taking these factors into consideration, we
divided all image samples into six categories, including "Good quality", "Raindrops",
"Middle shadow", "Below shadow", "Water reflection", and "Dark" (**Fig. 2**). "Good
quality" contains image samples without obvious noise or shadow. All the other images
lose some feature information due to noise, shadows, reflections, or dim lighting. To
ensure the model performance under different environmental conditions, we designed
an automated categorization procedure (**Fig. 3**) to screen the raw images and exclude
the "Raindrops" and "Dark" samples from model training.

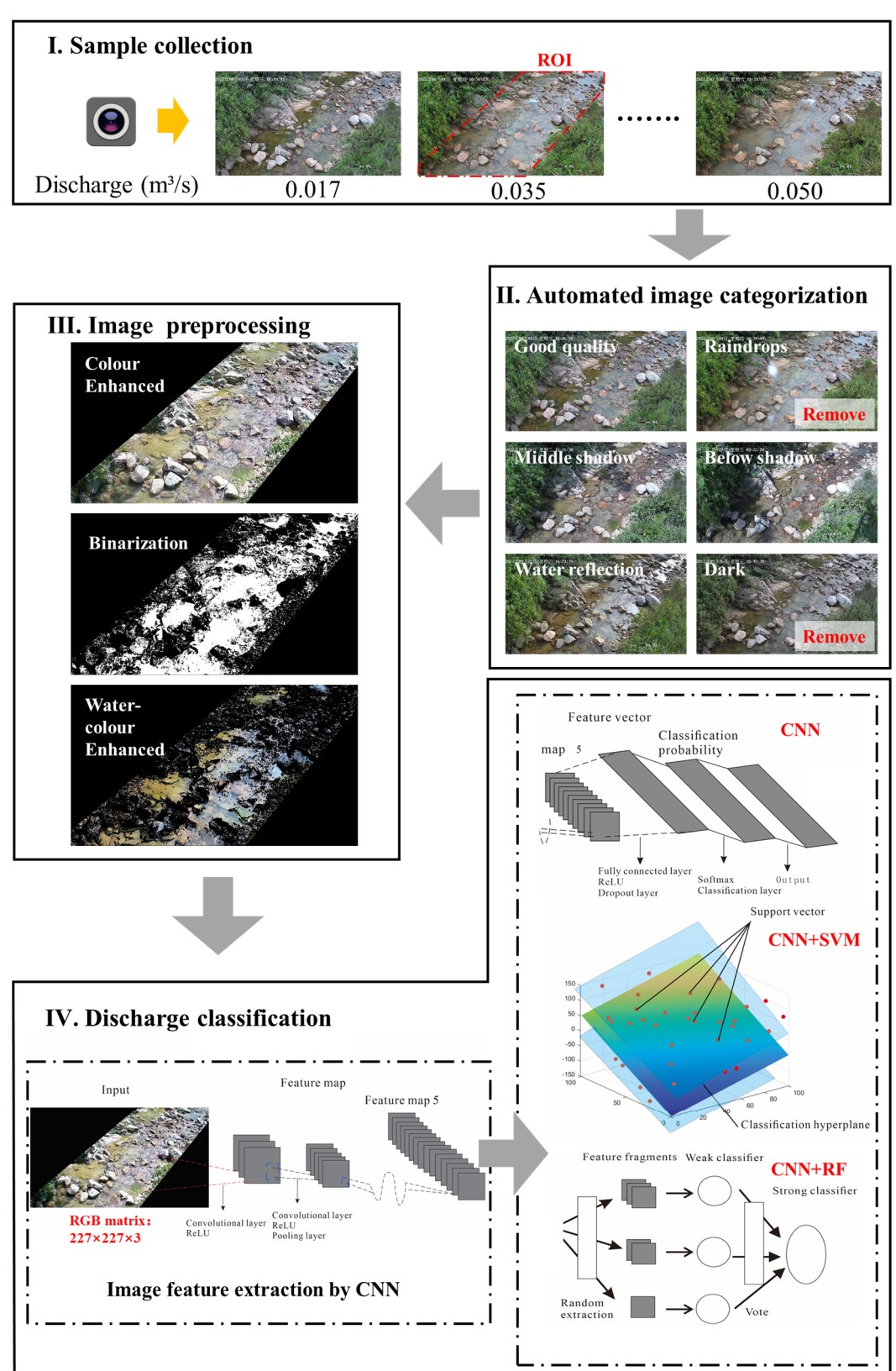


**Figure 2.** Flowchart of image processing and discharge monitoring.


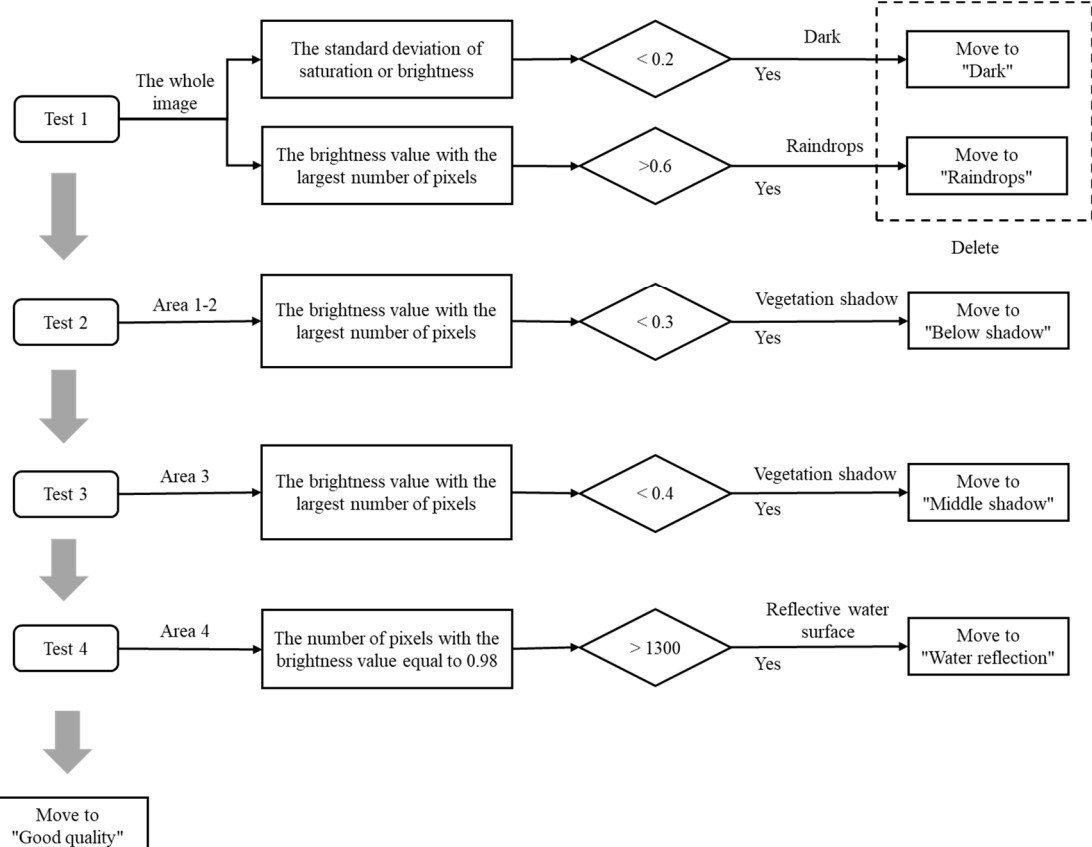


**Figure 3.** Procedure of automated image categorization.

Firstly, we selected four areas in the image as the detection areas (**Fig. 4a**) where the
special conditions mentioned above commonly occurred: the upper and lower shadows
in the target stream section mainly appeared in Area 3 and Area 1&2, respectively;
disturbance of water surface reflection was mostly found in Area 4. Then, the thresholds
of saturation or brightness in the four detection areas for image categorization were
determined manually by comparing image samples under different conditions. The
four-step procedure includes: (1) "Dark" images (**Fig. 4f-2**) were identified when the
standard deviation of the brightness or saturation of the full image was less than 0.2. (2)
"Raindrops" images (**Fig. 4f-3**) were identified when the brightness of the whole image
with the largest number of pixels was greater than 0.6. These two types of images were
excluded from the training samples. (3) "Below shadow" (**Fig. 4b-2**; **Fig. 4c-2)** and
"Middle shadow" images (**Fig. 4d-2**) were identified when the brightness value with
the largest number of pixels in Area 1&2 and Area 3 was less than 0.3 and 0.4,
respectively. (4) "Water reflection" images were identified when the number of pixels
with a brightness value of 0.98 in Area 4 exceeded 1300 (**Fig. 4e-2**). The images passing
all the tests in the procedure were considered "Good quality" samples. The other charts
in **Fig. 4** show the saturation and brightness distributions derived from a typical "Good
quality" image.

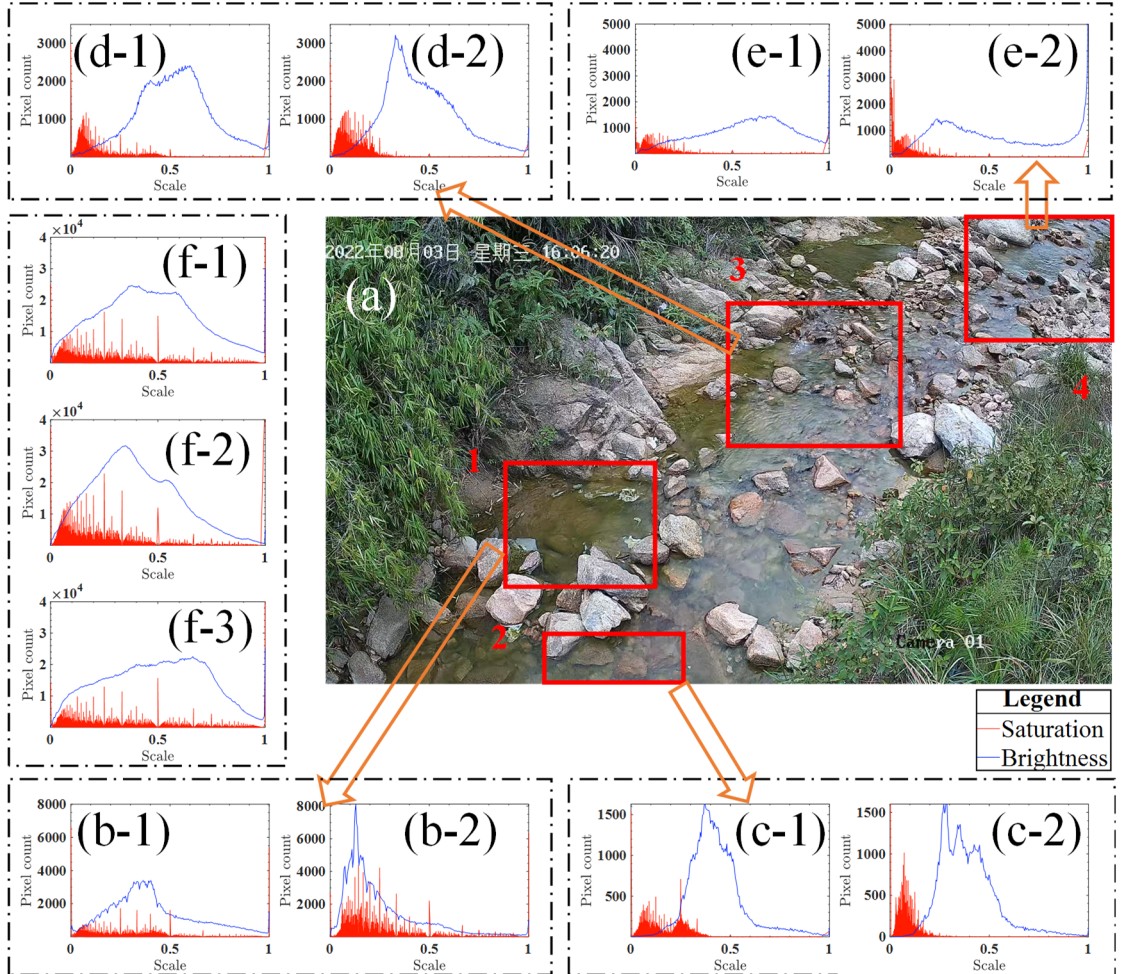


**Figure 4.** Comparison of saturation and brightness distributions in the four detection areas under different environmental conditions. The horizontal axis is the interval range (0-1) of saturation and brightness in HSB space. The vertical axis indicates the number of pixels under a certain saturation or brightness value. Figures b-1, c-1, d-1, and e-1 display the saturation and brightness distributions in Area 1-4 of a "Good quality" sample. Figures b-2, c-2, d-2, and e-2 display the results derived from samples of "Below shadow" (b-2; c-2), "Middle shadow" (d-2), and "Water reflection" (e-2), respectively. Figures f-1, f-2, and f-3 display the saturation and brightness distributions of an entire image, derived from "Good quality", "Dark", and "Raindrops" samples,

respectively.

**2.3.2 Color enhancement**
In order to highlight the stream features embedded in the images and avoid image
information redundancy, we compared three commonly used color enhancement
approaches to process the image samples.
**(1) Color Enhanced.** A dynamic histogram equalization technique (Abdullah-Al-
Wadud et al., 2007; Cheng and Shi, 2004) was used to enhance contrast and emphasize
stream features. First, vegetation areas on both sides of the stream were cropped and
filled with black. Then, histogram equalization was used to enhance the contrast
between light and dark, i.e., brighten the bubbles, swirls, ripples, splashes, water
coverage, etc., and darken the bottom stones and reflections in the water.
**(2) Binarization.** Binarization of image information can decrease the
computational load and enable the utilization of simplified methods compared to 256
levels of grey-scale or RGB color information (Finlayson et al., 2002; Sauvola and
Pietikäinen, 2000). In this case, the RGB and HSB (Hue, Saturation, Brightness)
information extracted from images suggests that the brightness of the stream water
under daylight ranges from 0.2 to 0.7, and the values of three color components follow:
$$R(x, y) + G(x, y) + B(x, y) > 350 \qquad (2)$$
Where $R(x, y)$, $G(x, y)$ and $B(x, y)$ respectively represent the red, green, and blue
color values of the pixel (x, y). The original image was transformed into a binary image
by assigning the values of "1" and "0" to the pixels within and out of the water body,
respectively.

**(3) Water-color Enhanced.** Considering that water-color features may carry some

useful information on discharge (Kim et al., 2019), we tested a new pre-processing
method combining the two approaches above. The RGB information of the original
image within the water body areas was kept unchanged, while the non-water body areas
were filled with black color. Then, the water body areas were further enhanced with the
histogram equalization method to highlight the edge transition between the water body
and the background (Abdullah-Al-Wadud et al., 2007).
**2.3.3 Image denoising**
Images pre-processed by all of three approaches still contain large amounts of noise
due to environmental disturbances and edge oversharpening caused by image contrast
enhancement (Herzog et al., 2022). Therefore, the wavelet transform (Zhang, 2019)was
adopted to denoise the image samples. We chose a compromise threshold between hard
and soft thresholds as the threshold function (Chang et al., 2010). When the wavelet
coefficient is greater than or equal to the threshold, a compromise coefficient $\alpha$ ranging
from 0 to 1 is added before the threshold to achieve a smooth transition from hard to
soft thresholds, as
$$\lambda = \frac{median(d_j(k))}{0.6745} \times \sqrt{2\log(M \times N)} \qquad (3)$$
$$\omega_\lambda = \begin{cases} [sign(\omega)](|\omega| - \alpha\lambda), |\omega| \geq \lambda \\ 0, |\omega| \geq \lambda \end{cases} \qquad (4)$$
where $j$ is the scale of wavelet decomposition, $d_j(k)$ is the coefficient of wavelet
decomposition, *M* and *N* are the length and width of images, $\omega$ is the wavelet
coefficient, $\lambda$ is the set threshold, and $sign$ is the sign function. In this case, $M\times$
$N$=2560×1440, $\alpha$=0.5.

**2.4 Correlation between color information and discharge**

The unstructured image data of mountain streams implicitly contains many stream
features relevant to discharge, such as the width and depth of streams, the coverage of
water surface, and spatial distributions of flow direction and flow velocity. In this study,
we attempted to achieve discharge monitoring by establishing empirical relationships
between the RGB color information of the water body and the discharge volumes. We
first explored the correlation between the combination of R/G/B values ($a\bar{R} + b\bar{G} +$
$c\bar{B}$, where $\bar{R}$, $\bar{G}$, $\bar{B}$ are the mean values of red, green and blue channels of an image,
respectively, and *a*, *b*, and *c* are coefficients to be determined) in the region of interest
(ROI, see **Fig. 2**) and the discharge conditions. Spearman's rank correlation coefficient
between $a\bar{R} + b\bar{G} + c\bar{B}$ and discharge is calculated as

$$r_s = 1 - \frac{6\sum_{i=1}^{n} d_i^2}{n(n^2-1)} \tag{5}$$

where *n* is the number of samples, $d_i$ is the difference between the ranks of R/G/B values
and discharge of each image sample.

**2.5 Algorithms of discharge estimation**

We used three algorithms to establish discharge classification models (**Fig. 2**), including
convolutional neural network (CNN), support vector machine (SVM), and random
forest (RF). The data of the RGB color matrix derived from pre-processed images was
used as model inputs. SVM and RF were coupled with CNN to explore the potential
merits of traditional machine learning algorithms in improving the classification
accuracy and efficiency of CNN-based discharge classifiers. All the embedding image
features are normalized and regularized before passed to classifiers to avoid overfitting
for CNN-based models.
**2.5.1 Convolutional Neural Network (CNN)**
Deep convolutional neural network allows computational models composed of multiple
processing layers to learn representations of data with multiple levels of abstraction,
which have brought breakthroughs in processing images, video, speech, and audio
(Lecun et al., 2015). The AlexNet architecture (Krizhevsky et al., 2017) was used to
construct our model. Parameters of the semantic layer of the model were calibrated to
achieve feature extraction and classification of stream images. The image size was first
rescaled from 2560×1440 to 227×227 to facilitate the migration of trained AlexNet. A
227×227×3 (length×width×color) matrix was retrieved from each image as the model
input. There were five built-in convolutional layers, using a 3×3 convolution kernel and
a 3×3 pooled kernel. We replaced the last three layers of AlexNet with a full-connection
layer, a softmax layer, and a classification layer, leaving all other layers intact. The
parameters of the full-connection layer were set according to the number of selected
discharge values. The ReLU function was used as the convolutional layer activation
function to extract and pass on the water coverage features. The SoftMax function was
the activation function of the output layer, and the extracted feature vectors were
compressed under each discharge label. The probability that a stream image falls into a
discharge label was calculated as

$$P(y|x) = \frac{e^{h(x,y_i)}}{\sum_{i=1}^{n} e^{h(x,y_i)}} \tag{6}$$

where $x$ is the feature vector extracted by CNN, $y$ is the discharge label, $n$ is the number
of labels, $h(x, y_i)$ is the linear connectivity function. The training method for CNN
was stochastic gradient descent with momentum, with 15 samples in small batches, a
maximum number of rounds of 10, a validation frequency of 3 epochs, and an initial
learning rate of 0.00005. The samples were shuffled in every epoch. The loss function
for discharge classification was Cross-Entropy Loss, as

$$L = -\frac{1}{N} \sum_{i=1}^{N} \sum_{c=1}^{C} y_{i,c} \log(p_{i,c}) \tag{7}$$

where $L$ is the value of loss, $N$ is the number of samples, $C$ is the number of discharge
classes, $y_{i,c}$ represents the value of the true label for the $i^{th}$ sample in the $c^{th}$ class using
one-hot encoding, and $p_{i,c}$ represents the probability of $i^{th}$ sample belonging to $c^{th}$
class calculated by CNN.

**2.5.2 Convolutional Neural Network coupled with Support Vector Machine**
**(CNN+SVM)**
SVM is a machine learning method based on structural risk minimization and Vapnik–
Chervonenkis (VC) dimension theory (Cortes and Vapnik, 1995). It has been widely
used in image processing, pattern recognition, fault diagnosis, prediction and
classification (Burges, 1998), which can help to capture key samples and eliminate
redundant samples by finding the optimal hyperplane. Compared with neural networks,
which rely on large training samples and tend to fall into local optima, SVM can achieve
global optima with a simpler model structure (Hanczar et al., 2010; Matykiewicz and
Pestian, 2012). However, the SVM-based classifier requires manual input of image
features. Therefore, we coupled CNN and SVM to achieve automatic discharge
classification. Image features extracted by CNN (i.e., the output of the $5^{th}$ CNN pooling
layer) were fed into SVM classifiers to calculate discharge. The extracted image
features, coded with a "one-vs-all" scheme, were used to train binary SVM classifiers.
Specifically, one SVM classifier with a linear kernel function was trained for each
discharge class to distinguish that class from the rest. The hinge loss function was
employed to optimize the entire model by maximizing the margin between discharge
classes.

**2.4.3 Convolutional Neural Network coupled with Random Forest (CNN+RF)**
RF (Tin Kam, 1995) is a flexible machine-learning algorithm that combines the output
of multiple decision trees to reach a single result. Each decision tree depends on the
values of a random vector sampled independently and with the same distribution for all
trees in the forest (Breiman, 2001; Panda et al., 2009). It is an integrated algorithm of
the Bagging type (Aslam et al., 2007) that combines multiple weaker classifiers, and
the final result is obtained by voting or averaging to improve accuracy and
generalization performance. We here used an RF comprising 350 decision trees and five
decision leaves for discharge calculation. The coupling method of CNN+RF mirrors
that of CNN+SVM, using the same pooling outputs of CNN as inputs for RF discharge
classifier. RF is trained to assign optimal weights to each decision tree and leaf without
a specific loss function.

**2.6 Model evaluation metrics**
The performance of discharge classification models was measured by four widely used
metrics, including classification accuracy, F1 score, coefficient of determination ($R^2$),
and root mean square error (RMSE).
(1) Accuracy:
$$Accuracy = \frac{\sum_{i=1}^{k} TP_i}{N} \tag{8}$$
where $TP_i$ is the number of correctly classified samples in the $i^{th}$ discharge class; $N$ is
the total number of samples; $k$ is the number of discharge classes.
(2) F1 score:
$$F1 = \frac{2 \times Precision \times Recall}{Precision + Recall} \tag{9}$$
where *Precision* is the ratio of true positive classification ($TP_i$) to the sum of $TP_i$ and
the number of misclassified samples with the $i^{th}$ discharge simulated by a model ($FP_i$);
*Recall* is the ratio of $TP_i$ to the sum of $TP_i$ and the number of misclassified samples
with the observed $i^{th}$ discharge ($FN_i$), calculated as
$$Precision = \sum_{i=1}^{k} \frac{n_i}{N} \times \frac{TP_i}{TP_i + FP_i} \tag{10}$$

$$Recall = \sum_{i=1}^{k} \frac{n_i}{N} \times \frac{TP_i}{TP_i + FN_i} \tag{11}$$

where $n_i$ is the number of samples that fall in the $i^{th}$ class.

(3) R²

$$R^2 = 1 - \frac{\sum_{j=1}^{N}(y_j - \hat{y}_j)^2}{\sum_{j=1}^{N}(y_j - Y)^2} \tag{12}$$

where $y_j$ and $\hat{y}_j$ are the observed and simulated discharge, respectively; $Y$ is the mean

discharge.

(4) RMSE

$$RMSE = \sqrt{\frac{1}{N}\sum_{j=1}^{N}(y_j - \hat{y}_j)^2} \tag{13}$$

# 3 Results

## 3.1 Correlation analysis

We first performed a preliminary correlation analysis between the RGB matrices in ROI

and the discharge values. Traversing the common algebraic combinations of the three

colors, we found that $-\bar{R} + 7.5\bar{G} - 6.5\bar{B}$ ($\bar{R}$, $\bar{G}$, $\bar{B}$ are the mean values of red, green

and blue channels of an image, respectively) had a spearman correlation coefficient of

0.67 with discharge (p-value < 0.01), indicating that the discharge is significantly

correlated with the color combination value at the 99% confidence level (**Fig. 5**). Such

result suggests that discharge conditions are embedded in RGB information of

mountain streams to some extent, which could be further retrieved and refined by CNN

models.

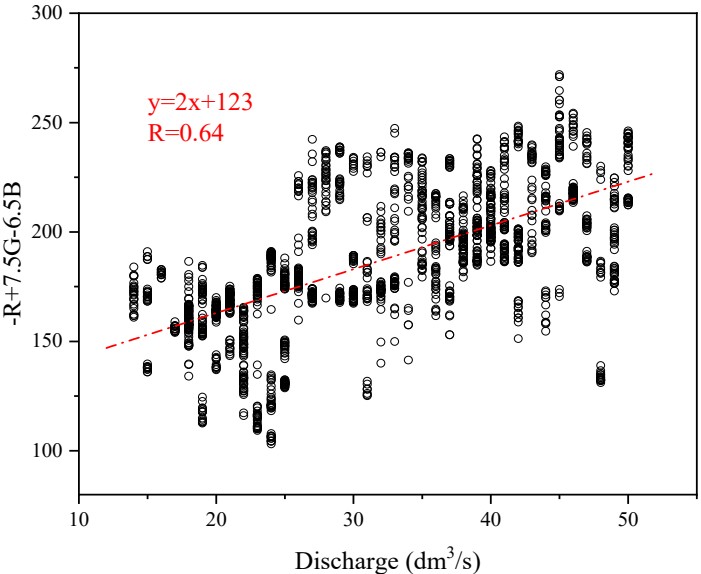

**Figure 5.** Correlation between RGB color values and corresponding discharges.

## 3.2 Effectiveness of automated image categorization

Most of the previous image-based studies only selected unblemished images for discharge or velocity monitoring, which resulted in poor model performance under environmental disturbances (Leduc et al., 2018; Chapman et al., 2020; Herzog et al., 2022). In this study, we also included samples under the influence of vegetation shadows and water reflection for model training. We selected approximately 100 stream images corresponding to each discharge volume (at the interval of 0.001 m³/s) from the pre-processed samples (3168 images in total). The databases of "Good quality", "Middle shadow", "Below shadow", and "Water reflection" were approximately sampled in the ratio of 7:0.6:1.4:1 (2146:244:437:341 images) to ensure the representation of different environmental conditions. The samples were distributed evenly in each discharge interval to avoid bias towards particular discharge conditions

and enhance model performance on high and low flows (Wang et al., 2023).
**Fig. 6** demonstrates the difference in classification accuracy of monitoring discharge
by the defective images, using two sets of models trained with only "Good quality"
images and samples filtered by automated image categorization, respectively. Results
derived from the three discharge classification models and three color-enhancing
methods consistently suggest that the procedure of automated image categorization can
significantly improve model performance in apprehending defective images.
Classification accuracy of the models trained with only "Good quality" samples
staggered between 11.8%-18.7%, while the accuracy of the models trained after
automated image categorization was higher than 79.0% (79.0%-97.4%) regardless of
the choices of color processing method and deep learning model. The average
difference in classification accuracy between the two sets of training samples reached
73.9%. The proportionate inclusion of defective images with vegetation shadow and
water surface reflection enhances the anti-interference ability of the models in complex
environments.

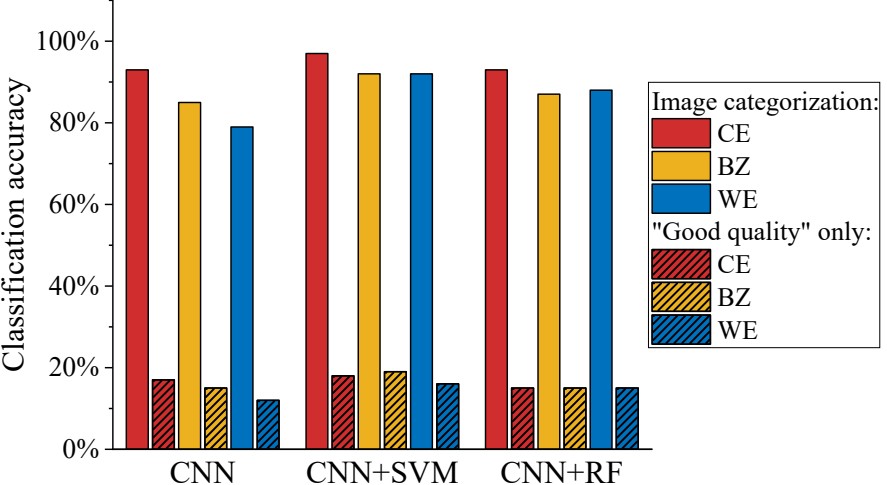


**Figure 6.** Accuracy of discharge classification of images under environmental

disturbances. Bars with and without patterns show the results using the models trained
with only "Good quality" samples and samples after automated image categorization,
respectively. Color enhancement methods include Color Enhanced (CE), Binarization
(BZ), and Water-color Enhanced (WE).

## 3.3 Model training and validation

After the treatments of color-enhancing, image denoising, and automated image
categorization, the images were randomly divided into training and validation sets by
the ratio of 7:3, and then used for model training and validation, respectively.

### 3.3.1 Loss changes

The changes in training and validation loss of the CNN models driven by three types of
color-enhanced images are demonstrated in **Fig. 7**. In the initial twenty epochs, the
training loss values decreased rapidly from 7.70 to 3.73 (Color Enhanced), from 5.91
to 3.73 (Binarization), and from 5.41 to 3.80 (Water-color Enhanced), respectively.
Subsequently, the decreasing rates slowed during the following 1000 epochs, averaging
around -0.0027 to -0.0030 per epoch. The loss value usually stabilizes after 1000 epochs
in CNN training (Keskar et al., 2016). In our case, the loss value began to flatten after
the 1300[th] epoch, signifying convergence towards a consistent loss value below 1.00
across all three color-enhancing methods. Therefore, we set the maximum training
epochs to 1470 to ensure model performance while avoiding overfitting.
The proximity between the training and validation loss changes at the final few
epochs is an important indicator that the model is not suffering from overfitting. A
commonly acknowledged benchmark of such proximity is approximately 0.1 to 0.2
(Heaton, 2018). In our CNN models, the validation loss values at the final epoch were
0.60, 0.78, and 0.63, respectively, which were 0.19, 0.08, and 0.07 lower than the
corresponding training loss. Such results suggest that the models did not suffer from
overfitting or underfitting.

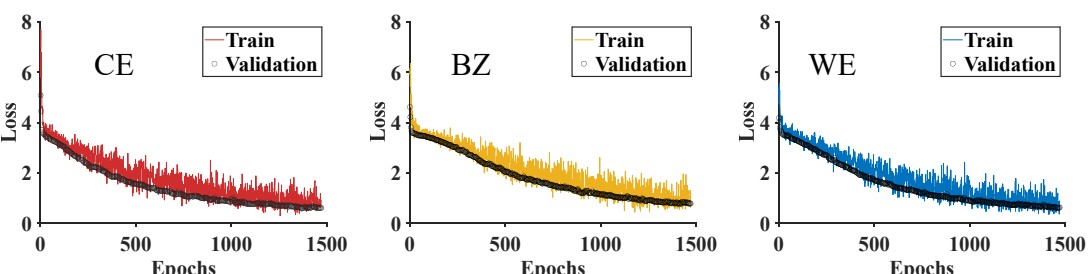

**Figure 7.** Changes in training and validation loss of the models driven by three types
of color-enhanced images. Color enhancement methods include Color Enhanced (CE),
Binarization (BZ), and Water-color Enhanced (WE).

### 3.3.2 Comparison of discharge classification models

The heap map (**Fig. 8**) visualizes the performance of different models in classifying the validation image set with three tested color-enhancing methods under different environmental conditions. Results show that all three models (i.e., CNN, CNN+SVM, CNN+RF) can achieve satisfactory performance on discharge classification. The $R^2$ under all environmental conditions was greater than 0.97, suggesting that the simulated discharge was significantly correlated to the flowmeters' measurement. The comparison of model performance generally shows consistency under different environmental conditions. Higher classification accuracy and F1 score are always accompanied by higher $R^2$ and lower RMSE, showing that CNN-based models perform well in accurately recognizing true discharge and handling outliers. Among the three models, CNN is more likely to over- or under-estimate discharge than both CNN+SVM and CNN+RF, with classification accuracy and F1 score 8.6~13.4% and 0.084~0.115 lower than CNN+SVM and CNN+RF, respectively. With all environmental conditions taken into account, CNN+SVM shows the best overall performance with the highest classification accuracy of 88.6%, the highest F1 score of 0.878, the highest $R^2$ of 0.989, and the lowest RMSE of 1.08 dm$^3$/s. Such results could be related to the size of our samples and the characteristics of the features extracted by deep layers of CNN. The features extracted from stream images under one specific flow discharge show similarities, which highlights the SVM's capability in classifying the embeddings from small samples with linear features.

### 3.3.3 Comparison of color-enhancing methods

Among the three tested color-enhancing methods, the Color Enhanced approach generally shows the best performance in discharge classification. Models driven by Color Enhanced images achieved higher classification accuracy (+2.3%~+7.4%), higher F1 score (+0.033~+0.067), higher $R^2$ (+0.001~+0.009), and lower RMSE (−0.068 ~ −0.415 dm$^3$/s) than those driven by images processed with Binarization and Water-color Enhanced. This is partly due to the different treatments in the edges of the water body. Binarization and Water-color Enhanced relatively cause larger deviation from the real edges, while Color Enhanced retains the image information to the maximum extent. Binarization reduces the cost of discharge computation and data storage by transforming raw stream images into binary images, and thus facilitates real-time monitoring by embedded end-to-end devices (e.g., mobile phones) with insufficient computing power (Shi et al., 2019). Considering that the color and texture of the water surface vary significantly with discharge volumes while the background is relatively stable, we proposed the Water-color Enhanced approach that only processes color information within the water body. In our experiment, it took only 0.0154s to recognize flow discharge from one Binarization image with an Intel (R) Core (TM) i7-10750H CPU, which was 36% and 22% faster than that of Color Enhanced and Water-color Enhanced images, respectively. Such results suggest that it is beneficial to retain the background information to the maximum extent and include the non-water parts of

mountain streams in image processing. However, future applications of image-based
discharge monitoring need to strike a balance between accuracy and speed when
choosing color processing methods.

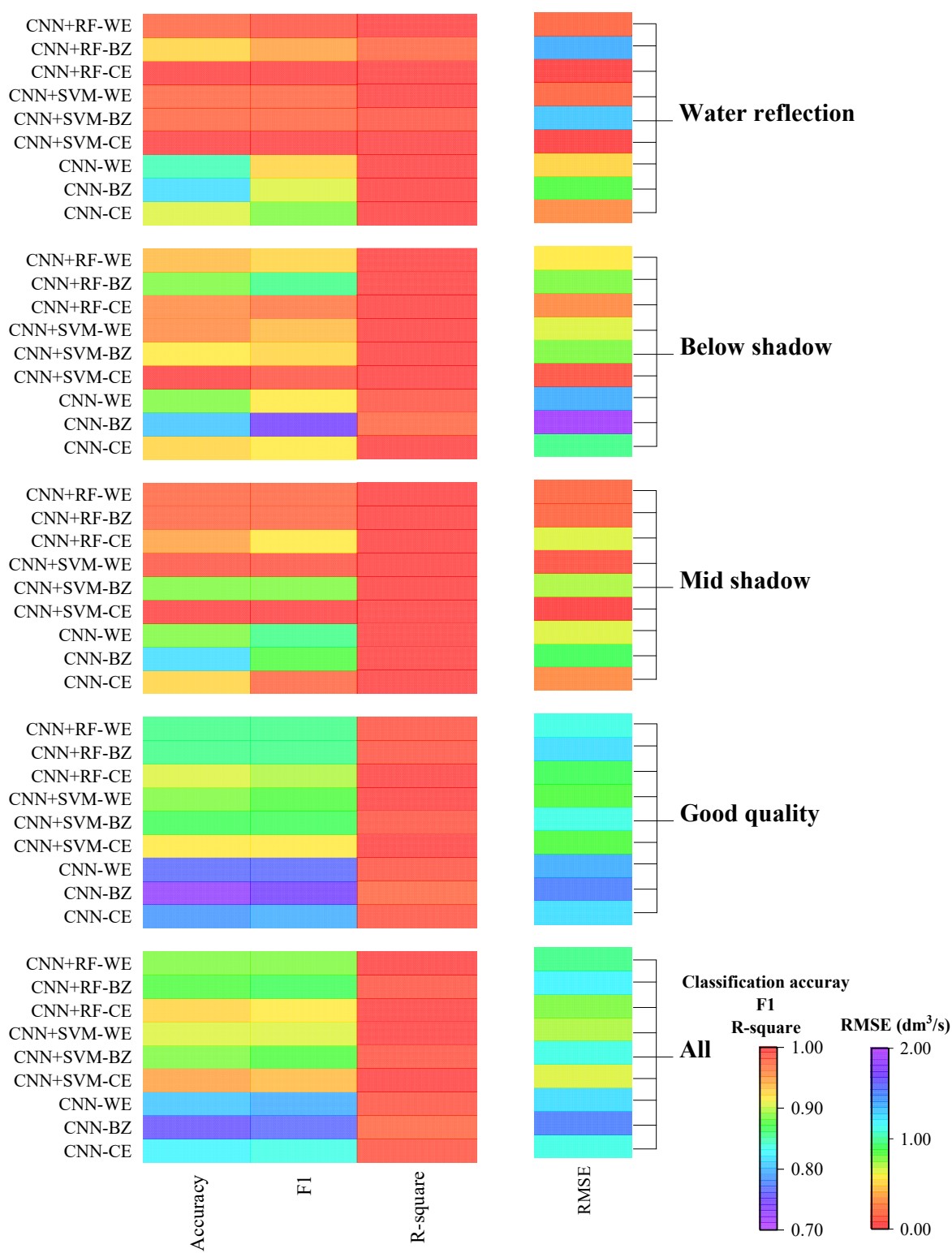


**Figure 8.** Performance of discharge classification models under different
environmental conditions. Color enhancement methods include Color Enhanced (CE),

Binarization (BZ), and Water-color Enhanced (WE).

## 4 Discussion

The existing image-based methods usually rely on either the estimations of flow
velocity and cross-section area or assumptions on stage-discharge correlation (Tauro et
al., 2017; Leduc et al., 2018; Davids et al., 2019; Li et al., 2019). The first type of
method uses image-derived surface velocity to estimate sub-sectional mean streamflow
velocity and spatial integration of discharge (Le Coz et al., 2012). The difficulties in
capturing cross-sectional characteristics and the relationship between flow velocity and
water depth limit their application in small mountain streams. The second type of
method retrieves river geometry directly through remote sensing, yet the accuracy is
primarily determined by the empirical assumptions on the relationships among water
depth, velocity, and discharge (Gleason and Smith, 2014; Young et al., 2015). In this
study, we proposed a new camera-based method to directly establish the relationship
between the RGB matrices of stream images and the classification probabilities of
discharge. The unique merit of the CNN-based model is its capability in automatically
extracting and refining discharge-related features from image samples, which improves
the accuracy and applicability of the model. Previous attempts suggest that the selection
of image features can significantly affect the performance on classification of stream
images (Tauro et al., 2014). For example, Chapman et al. (2020) manually extracted
features from pre- and post-weir images and used them as the inputs of machine
learning models. However, the dominant image features relating to stream discharge
could vary across different environments (e.g., topography, vegetation on river banks,
water quality), limiting the transferability of such manually identified features.
Weather conditions (e.g., sun position, fog, rain) are the most common difficulties
that reduce picture quality (Leduc et al., 2018). Therefore, we designed an automated
procedure for categorizing samples by their brightness and saturation: (a) select four
areas in the image as detection areas, (b) eliminate images with insufficient light or
raindrops on the lens, (c) identify thresholds and classify the remaining images into four
categories for further model training, including the images under the influence of
vegetation shadow and overexposure caused by water reflection in certain angles. Such
inclusion and categorization of defective samples have significantly enhanced the anti-
interference ability of the model, facilitating uninterrupted discharge monitoring
through the daytime. These factors and the thresholds of brightness and saturation are
site-specific and require manual trials to identify them. However, after adequate initial
calibration, an established model can be used for the same site for extended periods and
repeated installations of camera systems.
The training and validation of deep learning models require a large number of
representative samples (He et al., 2016). We collected a total of 7757 image samples
from July 20$^{th}$ to September 27$^{th}$, 2022, and 3168 images were used for model training
and validation after image screening and categorization. Although we executed an
effective automatic categorization procedure on the acquired image samples, it is
undeniable that the training and validation sets didn't cover all environmental
disturbances. For example, the time of sunrise and sunset, the appearance of water
surface reflections, and the coverage of vegetation shadows are affected by the angles
of sunlight and vary with seasons. With sufficient artificial lighting or installation of a
night-vision infrared camera (Royem et al., 2012), the images during nighttime can also
be used for discharge monitoring after training. More image samples are needed to
enrich the representativeness of the model in further studies. Another limitation is that
we have focused on low and average flow conditions in the model training due to the
lack of high-quality flood samples. In tropical and subtropical mountain streams of
southern China, floods usually occur during rainstorms and only last for a short time.
Heavy rainfalls constantly block the camera lens with raindrops, and the rapid
streamflow movement during heavy rainfall tends to cause blurred images, which can
only be partly improved by increasing the shutter speed and adjusting the camera
position. Moreover, site-specific field data is crucial for identifying the criteria for
image categorization and model training, which restricts the broader applicability of
our approach in ungauged basins, where such field data may not be readily available.
Further research on integrating multiple data sources and surveying approaches is
warranted for developing a more generalizable method.

## 545  5 Conclusions

This study presents a novel method for discharge monitoring of mountain streams using
deep learning techniques and a low-cost solar-powered commercial camera
(approximately \$200). The results confirmed our hypothesis that the discharge-relevant
stream features embedded in a large number of RGB images can be implicitly
recognized and retrieved by CNN to achieve continuous discharge monitoring.
Coupling CNN and traditional machine learning methods can potentially improve
model performance in discharge classification to various extents. In this case, the
classification accuracy, F1 score, and $R^2$ of CNN+SVM and CNN+RF were
9.1%~14.4%, 0.084~0.115, and 0.006~0.010 higher, respectively, while RMSE was
0.31~0.51 dm$^3$/s lower compared to CNN. Proper image pre-processing and
categorization can largely enhance the applicability of image-based discharge
monitoring. In an environment under complex disturbances such as mountain streams,
image quality is constantly interfered with by shadows of vegetation on the river banks.
The automated image categorization procedure can effectively recognize discharge
from defective images by filtering samples under different conditions and improve
model robustness. The comparison of the three color-enhancing approaches also
confirms the importance of including the non-water parts (e.g., large rocks) and
retaining the background information to the maximum extent in the image analysis.
The proposed method provides an inexpensive and flexible alternative apparatus for
continuous discharge monitoring at rocky upstream mountain streams, where it is
challenging to identify the cross-section shape or establish a stable stage-discharge
relationship. Site-specific field data is needed to identify the criteria for image
categorization and model validation. However, it circumvents the potential errors in

assuming cross-section characteristics, such as the relationship between water depth and flow velocity, and represents a new direction for applying deep learning techniques in acquiring high-frequency discharge data through image analysis.

## Code/Data availability

The code and data are available upon request from the corresponding author.

## Author contribution

KD and CF conceptualized the experiments. GY, ZZ, and QZ curated the data. All authors participated in the investigation. CF, GY, ZZ, and QZ wrote the original draft and visualized the data. KD reviewed and edited the final version of the manuscript.

## Competing interests

The authors declare no competing interests.

## Acknowledgments

This work was supported by the National Key Research and Development Program of China (2021YFC3200205, 2021YFC3001000), the National Natural Science Foundation of China (52379032), the Guangdong Basic and Applied Basic Research Foundation (2023A1515012241, 2023B1515040028), and the Guangdong Provincial Department of Science and Technology (2019ZT08G090).

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
