# Peer review of "1Technical Note: Monitoring discharge of mountain streams by2retrieving image features with deep learning"

_EGUsphere, 2023_

## Author Comment (AC1)

We thank the reviewer for the insightful and constructive comments. Please see the following point-to-point response.

(1)Please check the copyright of the image in Figure 2. It seems that you directly used the image of LeNET as the part of your neural network.

Re: Despite some similarities to the LeNet-5 architecture in Lecun et al (1998), we have redrawn the figure to show the structure and layers of AlexNet (i.e., First Layer: Conv1, kernel size=11×11×3, ReLU; Second Layer: Conv2, kernel size=5×5×96, ReLU, Max Pooling; ...; Last Layer: Conv5, kernel size=3×3×256, with two fully connected layers and one classification layer followed by). Therefore, we believe there is no copyright infringement.

(2)Please explain the practicality of your method. You chose to focus on monitoring discharge within the range of 0.014 and 0.050 m$^3$/s. However, for discharge monitoring at a larger scale, can your sample size meet the practical needs? How do you evaluate the impact of the difference between the discharge obtained from the empirical formula, which serves as the ground truth, and the actual flow discharge on the stability of your algorithm, it might from different distributions? How do you assess whether the model obtained based on this sample size has sufficient generalization ability and avoids overfitting to data you collected?

Re: We agree that the practicality and stability of deep learning models rely on the sizes of training samples, and the model parameters are usually site-specific and require manual trials to identify them (Wu et al., 2018; Isensee et al., 2021). In this case, the range of 0.014-0.050 m$^3$/s covers most on-site flow conditions, except for extreme flooding events. We have clarified such limitations in the last paragraph of the Discussion: "More image samples are needed to enrich the representativeness of the model in further studies. Another limitation is that we have focused on low and average flow conditions in the model training due to the lack of high-quality flood samples. In tropical and subtropical mountain streams of southern China, floods usually occur during rainstorms and only last for a short time. Heavy rainfalls constantly block the camera lens with raindrops, and the rapid streamflow movement during heavy rainfall tends to cause blurred images, which can only be partly improved by increasing the shutter speed and adjusting the camera position."

In the revised manuscript, we have added a section "**3.3.1 Loss changes**" to show the training processes of deep learning models and address the overfitting/underfitting problems: "The changes of training and validation loss of the CNN models driven by three types of color-enhanced images are demonstrated in **Fig. 7**. In the initial twenty epochs, the training loss values decreased rapidly from 7.70 to 3.73 (Color Enhanced), from 5.91 to 3.73 (Binarization), and from 5.41 to 3.80 (Water-color Enhanced), respectively. Subsequently, the decreasing rates slowed down during the following 1000 epochs, averaging around -0.0027 to -0.0030 per epoch. The loss value usually tends to stabilize after 1000 epochs in CNN training (Keskar et al., 2016). In our case, the loss value began to flatten out after the 1300$^{th}$ epoch, signifying convergence towards a consistent loss value below 1.00 across all three color-enhancing methods. Therefore, we set the maximum training epochs to 1470 to ensure model performance while avoiding overfitting.

The proximity between the training and validation loss changes at the final few epochs is an important indicator that the model is not suffering from overfitting. A commonly acknowledged benchmark of such proximity is within a range of approximately 0.1 to 0.2 (Heaton, 2018). In our CNN models, the validation loss values at the final epoch were 0.60, 0.78, and 0.63, respectively, which were 0.19, 0.08, and 0.07 lower than the corresponding training loss. Such results suggest that the models did not suffer from overfitting or underfitting."

(3)As you have employed an end-to-end approach, could you explain the reasons behind your choice of pre-processing? How does the inclusion or exclusion of these measures impact your results? It is worth considering that the chosen pre-processing methods might inadvertently remove essential features. In comparison to standard processing operations, what advantages does your approach offer?

Re: Besides the image denoising treatment, we compared the effectiveness of three color-enhancement approaches in highlighting the stream features embedded in the images. Among the three methods, "Color Enhanced" is widely used to augment the embedded RGB information in computer vision, which could improve the quality of images for CNNs. "Binarization" reduces the cost of computation and data storage, transforming raw stream images into binary images, which has been recommended for real-time monitoring. In our experiment, it took only 0.0154s to recognize flow discharge from one Binarization image with an Intel (R) Core (TM) i7-10750H CPU,

which was 36% faster than that of Color Enhanced images. However, we agree that Binarization also removes some useful information from raw images, causing the accuracy of discharge classification 4.6%-7.4% lower (**Fig. 8** in the revised manuscript). The "Water-color enhanced" approach was proposed to strike a balance between accuracy and speed by processing color information within water body only.

We have added a section "**3.3.3 Comparison of color-enhancing methods**" to clarify the impact and implications of different pre-processing methods: "Among the three tested color-enhancing methods, the Color Enhanced approach generally shows the best performance in discharge classification. Models driven by Color Enhanced images achieved higher accuracy (+2.3%~+7.4%), higher F1 score (+0.033~+0.067), and lower RMSE (–0.068 ~ –0.415 dm$^3$/s) than those driven by images processed with Binarization and Water-color Enhanced. This is partly due to the different treatments in the edges of the water body. Binarization and Water-color Enhanced relatively cause larger deviation from the real edges, while Color Enhanced retains the image information to the maximum extent. Binarization reduces the cost of computation and data storage by transforming raw stream images into binary images, and thus facilitates real-time monitoring by embedded end-to-end devices (e.g., mobile phones) with insufficient computing power. Considering that the color and texture of water surface vary significantly with discharge volumes while the background is relatively stable, we proposed the Water-color Enhanced approach that only processes color information within the water body. In our experiment, it took only 0.0154s to recognize flow discharge from one Binarization image with an Intel (R) Core (TM) i7-10750H CPU, which was 36% and 22% faster than that of Color Enhanced and Water-color Ehanced images, respectively. Such results suggest that it is beneficial to retain the background information to the maximum extent and include the non-water parts of mountain streams in image processing. However, future applications of image-based discharge monitoring need to strike a balance between accuracy and speed when choosing color processing methods."

(4)How is your loss function defined, is only nllloss? and please provide details of your training algorithm. Why did you choose AlexNet? What are the differences between AlexNet and other neural networks proposed in recent years?

Re: We have added details of loss function and training algorithm of the CNNs in **Section 2.5.1** of

the revised manuscript. The details of model training and the process of loss changes have also been supplemented in **Section 3.3.1.**

We chose AlexNet because it is a validated deep neural network with a relatively simple model structure and fewer parameters (Canziani et al., 2016). Our goal is to develop a method that can be easily implemented with cameras to achieve real-time discharge monitoring. We agree that some new networks, such as GoogleNet (Szegedy et al., 2015), VGG-16 (Simonyan and Zisserman, 2014), and ResNet-50 (He et al., 2016), have been proposed in recent years. However, compared to AlexNet, these networks usually use deeper layers and more complex structures to extract the bottom semantic features, requiring larger computing power consumption and memory footprint. For example, VGG-16 contains 16 layers and about 130 million parameters in total.

(5)The author used CNN+SVM. Why did you design such a combination? Specifically, when choosing a specific layer's CNN output as the input to SVM, why did you select that particular layer? If the intention was to leverage high-level features extracted by CNN, why not directly use structures like AutoEncoders to compress the input into low-dimensional latent variables and then perform SVM decomposition based on these latent variables? Are you assuming the SVM can learn a better mapping than the NN? and how does the limitation of the SVM's input dimensionality affect your performance? SVM is sensitive to the hyperparameters, how did you select the hyperparameters for SVM? and how you optimized then?

Re: The combination of CNN and SVM has been proven helpful for image classification in previous studies. For example, Ahlawat and Choudhary (2020) developed a hybrid CNN+SVM model that achieved a recognition accuracy of 99.28% over MNIST handwritten digits dataset; Sun et al. (2019) proposed a new CNN+SVM method to tackle the high computational cost in the classification of remote sensing images, and achieved lower loss value and better generalization in model training. In our experiment, CNN+SVM performs best among the three discharge classification models regardless of the three color-enhancing approaches. Besides, the average training time of CNN+SVM with an Intel (R) Core (TM) i7-10750H CPU was 0.21h, which was about 1.93h faster than CNN and 0.35h faster than CNN+RF. These results are consistent with the existing research.

AutoEncoder can be used to compress high-dimensional features and retain the key embeddings in a low-dimensional space (Baldi, 2011). However, some information is lost during the

dimensionality reduction process (Jia et al., 2022). The reconstructed low-dimensional features may not capture all the details and nuances present in the high-dimensional features, leading to potential inaccuracies or distortions. Taking them into consideration, we directly input the features extracted by CNN into SVM classifier to ensure the discharge-related features are accurate, rather than compressing the features first. We chose the 5th AlexNet pooling layer's output as the input to SVM, because this layer is the deepest layer in AlexNet, and the embeddings calculated by this layer are the most representative. In general, as the depth of a neural network increases, the features tend to become more effective for classification (Tao et al., 2018).

We agree SVM is sensitive to the hyperparameters and the generalization performance of SVM can be improved by adjusting them (Cortes and Vapnik, 1995). In this study, we used the SVM toolbox provided by MATLAB, and the hyperparameters were optimized with the sequential minimal optimization (SMO) algorithm.

(6)All abbreviations should be defined when they first appear in the article and the author should correct them in the manuscript.

Re: Thank you for your careful review and thoughtful reminder. We have checked all the abbreviations in the manuscript and corrected them when they first appear.

(7)The authors mention in line 124-125 that "real-time discharge is calculated at a time step of two minutes", but it is unclear how to match video images with a temporal resolution of 5 minutes.

Re: Images were extracted from the videos at 5-minute intervals to avoid excessive similarity between adjacent images, and the corresponding discharge value of each image was the average of flowmeter's measurement during each time step.

(8)The author designed an automated categorization procedure to screen the raw images and exclude the "Raindrops" and "Dark" samples from model training. The author defines "Good quality" contains image samples without obvious noise or shadow in line 151-152, but how does the algorithm implement the determination of it?

Re: We have redrawn the flowchart of the automated image categorization (**Fig. 3**) in our revised manuscript to clarify the determination of "Good quality" samples. The description of the four-test

procedure has been corrected in **Section 2.3.1**, Line 180-189, as:

"(1) "Dark" images (**Fig. 4f-2**) were identified when the standard deviation of the brightness or saturation of the full image was less than 0.2. (2) "Raindrops" images (**Fig. 4f-3**) were identified when the brightness of the whole image with the largest number of pixels was greater than 0.6. These two types of images were excluded from the training samples. (3) "Below shadow" (**Fig. 4b-2**; **Fig. 4c-2**) and "Middle shadow" images (**Fig. 4d-2**) were identified when the brightness value with the largest number of pixels in Area 1&2 and Area 3 was less than 0.3 and 0.4, respectively. (4) "Water reflection" images were identified when the number of pixels with a brightness value of 0.98 in Area 4 exceeded 1300 (**Fig. 4e-2**). The images passing all the tests in the procedure were considered "Good quality" samples."

(9)In Figure 2, the image representing the dark images still seem to have relatively clear visibility, can the author's model effectively cover the 7:00-19:00 interval if such images are also directly removed? After removing the "Raindrops" and "Dark" samples, how many valid images remain?

Re: The "Dark" samples accounted for only a small fraction of the total dataset, comprising 231 out of 7757 images. After implementing automated image categorization and removing the "Dark" samples, we still have enough images covering the 7:00-19:00 interval (3168 valid images in total).

(11)The author built the model based on the idea of image classification, whereas discharge is a continuous variable more suitable for regression problems in deep learning. Therefore, the author's choice of the classification model needs to be justified.

Re: We agree that discharge is a continuous variable. However, the precision of discharge measurement is limited. In this study, discharge of mountain stream was classified at the interval of 0.001 m³/s. At such resolution, the outcomes of regression and classification models would be similar (Loh, 2011). We have clarified this in "**Section 2.2 Data**" of the revised manuscript, Line144-146: "A total of 7,757 image samples labeled with 37 discharge values between 0.014 and 0.050 m³/s at the interval of 0.001 m³/s were collected for model testing."

(12)During the process of constructing the model, the author needs to explain how the loss function was set and how the loss changed during training on both the training and validation sets.

Re: Following the reviewer's suggestion, we have made two revisions to the manuscript. Firstly, we have supplemented how the loss function of CNN (Cross-Entropy Loss) was set in **Section 2.5.1**. Secondly, we have recorded the loss value at each epoch during model training, and plotted the process of loss changes on both training and validation sets with three color-enhancing methods (**Fig. 7**).

(13)In line 135-136, the author mentions that "7,757 image samples labeled with 37 discharge values between 0.014 and 0.050 m³ /s were collected for model testing". However, in line 311, "100 stream images corresponding to each discharge volume for model training and validation" are used. Why only 100 stream images were used for model training and validation?

Re: Given the limited size of our dataset and the uneven distribution of stream images across different discharges, we selected approximately 100 stream images for each discharge value to ensure that the models would not be biased toward any particular discharge condition (Wang et al., 2023).

In the revised manuscript, we have rephrased the sentence to clarify it: "We selected approximately 100 stream images corresponding to each discharge volume (at the interval of 0.001 m³/s) from the pre-processed samples (3168 images in total). The databases of "Good quality", "Middle shadow", "Below shadow", and "Water reflection" were approximately sampled in the ratio of 7:0.6:1.4:1 (2146:244:437:341 images) to ensure the representation of different environmental conditions. The samples were distributed evenly in each discharge interval to avoid bias towards particular discharge conditions and enhance model performance on high and low flows (Wang et al., 2023)."

(14)Section 3.1 in the results was not mentioned before, and it should be reflected in the methods section.

Re: We thank the reviewer for the careful reminder. We have added a new subsection (**Section 2.4**) to introduce the purpose and method of analyzing correlation between color information and discharge.

(15)Coordinate labels in Figure 4 (b-2) are partly obscured.

Re: We have modified the coordinate labels in the figure.

(16)Figure 5 is more methodologically oriented, and it should be included in the methodology section rather than the results section.

Re: Following the reviewer's suggestion, we have moved the figure to Methods (**Section 2.3.1**).

(17)What is the significance of the author's additional classification of low shadow images, medium shadow images, and water reflection images? The author only mentioned the distribution of images in a ratio of 7:1:1:1 in lines 312-314.

Re: The significance of image categorization was discussed in two parts of the Results section. Usually, previous studies only selected unblemished images for discharge or velocity monitoring and excluded images with shadows in model training. Therefore, we first analyzed the effectiveness of image categorization in **Section 3.2** by comparing accuracy between two sets of models—one trained with only "Good quality" samples and the other with all the samples after image categorization. The results show that the procedure of automated image categorization can significantly improve model performance in apprehending defective images with shadow or reflection.

Second, in **Section 3.3** of the revised manuscript, we have redrawn the figure of model accuracy to show the results of different samples separately, namely the categories of "Good quality" "Middle shadow" "Below shadow" and "Water reflection".

(18)The authors have classified the video images, and in the results section, the effect of discharge recognition of "Good quality", "Below shadow", "Middle shadow", and "Water reflection" should be discussed individually.

Re: Following the reviewer's suggestion, we have added a more comprehensive analysis of models' performance on discharge classification using different categories of images in **Section 3.3.2 & 3.3.3**. The results (accuracy, F1 score, R-square, and RMSE) of image-based discharge simulation were analyzed for "Good quality", "Below shadow", "Middle shadow", and "Water reflection" samples, individually.

(19)Although the evaluation metrics used in the study are widely applicable, I suggest that the authors provide a brief introduction to the Spearman correlation coefficient, accuracy, RMSE and other metrics mentioned in the methodology section.

Re: Following the reviewer's suggestion, we have added an introduction to the evaluation metrics in **Section 2.4-2.6** of the revised manuscript.

(20)Under the condition of imbalanced data samples, the performance of the model cannot be comprehensively evaluated by the accuracy metric. It is recommended that the authors supplement the results section with comprehensive metrics such as F1 scores, which will help to improve the reliability of the research conclusions.

Re: Following the reviewer's suggestion, we have supplemented three commonly used metrics for model evaluation in the Results section, including F1 score, coefficient of determination ($R^2$), and root mean square error ($RMSE$). The definition and calculation of these metrics have been introduced in the Methods section.

**References:**

Ahlawat, S. and Choudhary, A.: Hybrid CNN-SVM Classifier for Handwritten Digit Recognition, Procedia Computer Science, 167, 2554-2560, 10.1016/j.procs.2020.03.309, 2020.

Baldi, P.: Autoencoders, unsupervised learning and deep architectures, Proceedings of the 2011 International Conference on Unsupervised and Transfer Learning workshop Washington, USA, 10.5555/3045796.3045801, 2011.

Canziani, A., Paszke, A., and Culurciello, E.: An Analysis of Deep Neural Network Models for Practical Applications, ArXiv, 7, 1-7, 10.48550/arXiv.1605.07678, 2016.

Cortes, C. and Vapnik, V.: Support-Vector Networks, Mach. Learn., 20, 273–297, 10.1023/a:1022627411411, 1995.

He, K., Zhang, X., Ren, S., and Sun, J.: Deep Residual Learning for Image Recognition, 2016 IEEE Conference on Computer Vision and Pattern Recognition (CVPR), Las Vegas, Nevada, 27-30 June 2016, 770-778, 10.1109/CVPR.2016.90, 2016.

Heaton, J.: Ian Goodfellow, Yoshua Bengio, and Aaron Courville: Deep learning: The MIT Press,

Genetic programming and evolvable machines, 19, 305-307, 10.1007/s10710-017-9314-z, 2018.

Isensee, F., Jaeger, P. F., Kohl, S. A. A., Petersen, J., and Maier-Hein, K. H.: nnU-Net: a self-configuring method for deep learning-based biomedical image segmentation, Nature Methods, 18, 203-211, 10.1038/s41592-020-01008-z, 2021.

Jia, W., Sun, M., Lian, J., and Hou, S.: Feature dimensionality reduction: a review, Complex & Intelligent Systems, 8, 2663-2693, 10.1007/s40747-021-00637-x, 2022.

Keskar, N., Mudigere, D., Nocedal, J., Smelyanskiy, M., and Tang, P.: On Large-Batch Training for Deep Learning: Generalization Gap and Sharp Minima, arXiv preprint arXiv:1609.04836, 10.48550/arXiv.1609.04836, 2016.

Lecun, Y., Bottou, L., Bengio, Y., and Haffner, P.: Gradient-Based Learning Applied to Document Recognition, Proceedings of the IEEE, 86, 2278-2324, 10.1109/5.726791, 1998.

Loh, W. Y.: Classification and regression trees, Wiley interdisciplinary reviews: data mining and knowledge discovery, 1, 14-23, 10.1002/widm.8, 2011.

Simonyan, K. and Zisserman, A.: Very Deep Convolutional Networks for Large-Scale Image Recognition, 75 pp., 10.48550/arXiv.1409.1556, 2014.

Sun, X., Liu, L., Li, C., Yin, J., Zhao, J., and Si, W.: Classification for Remote Sensing Data With Improved CNN-SVM Method, IEEE Access, 7, 164507-164516, 10.1109/ACCESS.2019.2952946, 2019.

Szegedy, C., Wei, L., Yangqing, J., Sermanet, P., Reed, S., Anguelov, D., Erhan, D., Vanhoucke, V., and Rabinovich, A.: Going deeper with convolutions, 2015 IEEE Conference on Computer Vision and Pattern Recognition (CVPR), Boston, MA, USA, 7-12 June 2015, 1-9, 10.1109/CVPR.2015.7298594, 2015.

Tao, Y., Xu, M., Lu, Z., and Zhong, Y.: DenseNet-Based Depth-Width Double Reinforced Deep Learning Neural Network for High-Resolution Remote Sensing Image Per-Pixel Classification, Remote. Sens., 10, 779, 10.3390/rs10050779, 2018.

Wang, R., Chaudhari, P., and Davatzikos, C.: Bias in machine learning models can be significantly mitigated by careful training: Evidence from neuroimaging studies, Proceedings of the National Academy of Sciences, 120, e2211613120, 10.1073/pnas.2211613120, 2023.

Wu, J., Yin, X., and Xiao, H.: Seeing permeability from images: fast prediction with convolutional

neural networks, Science Bulletin, 63, 1215-1222, 10.1016/j.scib.2018.08.006, 2018.

---

## Author Comment (AC2)

We are grateful to the reviewer's insightful and constructive comments. Please see the following point-to-point response.

(1)The discharge is usually gauged at a regular and stable channel. The channel the camera set is irregular and there are lots of rocks scattered in the channel. It makes the flow turbulent and unstable. I don't think it is a suitable location for monitoring discharge. The method of retrieving image features for discharge monitoring could distinguish the magnitude of discharge qualitatively, but it seems impossible to conduct quantitatively without labeled discharge from a gauge. It means the method is hard to be applied independently, especially at an irregular channel without stable stage-discharge relationship.

Re: We agree that it is challenging to monitor mountain discharge at upstream irregular channels due to the steep terrain and inaccessibility for field personnel. As the reviewer pointed out, most of the streamflow gauges are still set at downstream river channels nowadays. Therefore, we attempted to propose a method of monitoring the discharge of mountain streams by retrieving image features with deep learning models. Particularly, we tested two hypotheses: (1) the features of mountain streams (e.g., coverage of water surface, flow direction, flow velocity) embedded in RGB images can be recognized by suitable deep learning approaches to achieve effective discharge monitoring, and (2) proper image pre-processing and categorization can improve accuracy of image-based discharge monitoring of mountain streams. The proposed method provides a possible alternative apparatus for continuous discharge monitoring at rocky upstream mountain streams, where it is challenging to identify the cross-section shape or establish a stable stage-discharge relationship. Site-specific field data is still needed to identify the criteria for image categorization and model validation. The reviewer has pointed out an important direction for future research. Further efforts are needed to develop a method that can be transferred directly to other sites without model calibration. However, we believe that we have achieved our goal in this study despite the method's limitations.

(2) With the proposed method, it may be possible to generate more accurate estimation of discharge compared to large-scale particle image velocimetry (LSPIV) and particle tracking velocimetry (PTV) at a regular channel. I suggest a comparison among the these methods to make it more convincing.

Re: LSPIV and PTV have been widely used to acquire the flow velocity distribution on water surface in recent years. We indeed have tried these two methods when we started this study. Unfortunately, we found LSPIV and PTV unsuitable for monitoring rocky mountain streams due to the steep terrain, shallow water depth, and complex cross-section. As shown in the figure below (**Fig. 1**), the velocity distribution of the stream derived from PIV (Tauro et al., 2018; Thielicke and Sonntag, 2021) is irregular and chaotic, making it hard to calculate the average flow velocity at a certain cross-section, and thus discharge.

[Figure]

**Figure 1.** The velocity distribution at the study site at 14:00 pm 2022/9/10 calculated by PIV. The regions filled with red crosses represent streambank and the large rocks lying in the middle of stream. The arrows represent the flow directions, and the color represents the flow velocity.

(3) It is better to give a brief description of the advances of retrieving image features with deep learning in the section of introduction.

Re: Following the reviewer's suggestion, we have added a paragraph in Introduction to describe the advances of retrieving image features with deep learning, as:

"Unlike PIV and PTV, deep learning models possess the capability to extract discharge-related features from images of rivers or streams automatically. These models are able to adjust the weights assigned to each feature, eliminating the need for manual attention and reducing the risk of overemphasizing or misinterpreting features that are unresponsive to flow discharge (Canziani et al., 2016). Besides, deep learning models can extract low-level image features, such as edges,

textures, and colors (Jiang et al., 2021). These merits could be essential in retrieving information from images of mountain streams, particularly in regions with intricate cross-sectional profiles.*"*

(4) Lines 262-268. It is better to put these sentences in the section of introduction or methods.

Re: Following the reviewer's suggestion, we have added a new subsection (**Section 2.4**) to introduce the purpose and method of analyzing correlation between color information and discharge.

(5) Figure 4. The differences among the sub-figures, like b-1 and b-2 should be pointed out.

Re: We have added a detailed description in the caption of **Fig. 4**:

"Figures b-1, c-1, d-1, and e-1 display the saturation and brightness distributions in Area 1-4 of a "Good quality" sample. Figures b-2, c-2, d-2, and e-2 display the results derived from samples of "Below shadow" (b-2; c-2), "Middle shadow" (d-2), and "Water reflection" (e-2), respectively. Figures f-1, f-2, and f-3 display the saturation and brightness distributions of an entire image, derived from "Good quality", "Dark", and "Raindrops" samples, respectively."

(6) Lines 282-299. There are several thresholds in the section, such as values of 0.3 and 0.4 in line 295, 0.98 and 1300 in lines 297. How these values were determined.

Re: The thresholds for automated image categorization were determined manually by comparing image samples under different environmental conditions. Similar patterns were found in the saturation and brightness distributions of different categories of images. For example, the "Middle shadow" images showed the brightness values with the largest number of pixels in Area 3 were less than 0.4, while that of the other categories of image were higher than 0.4, so we chose 0.4 as the threshold to classify "Middle shadow" images in test 3. We have clarified this in the revised manuscript.

**References:**

Canziani, A., Paszke, A., and Culurciello, E.: An Analysis of Deep Neural Network Models for

Practical Applications, ArXiv, abs/1605.07678, 7-14, 10.48550/arXiv.1605.07678, 2016.

Jiang, P. T., Zhang, C. B., Hou, Q., Cheng, M. M., and Wei, Y.: LayerCAM: Exploring Hierarchical Class Activation Maps for Localization, IEEE Transactions on Image Processing, 30, 5875-5888, 10.1109/TIP.2021.3089943, 2021.

Tauro, F., Petroselli, A., and Grimaldi, S.: Optical sensing for stream flow observations: A review, Journal of Agricultural Engineering, 49, 199-206, 10.4081/jae.2018.836, 2018.

Thielicke, W. and Sonntag, R.: Particle Image Velocimetry for MATLAB: Accuracy and enhanced algorithms in PIVlab, Journal of Open Research Software, 9, 12-26, 10.5334/jors.334, 2021.

---

## Referee Report (RR1)

This study introduced a method for discharge monitoring of mountain streams using deep learning and a low-cost solar-powered commercial camera. Discharge monitoring at rocky upstream mountain streams has been a difficult task for a long time due to the complex topography. Although the image-based method was only tested at one single site, it provides a different idea that could serve as an alternative apparatus, or integrated into traditional approaches to improve data quality. The paper tackled a few important issues in streamflow image processing, including the treatment of images affected by the disturbances of water reflection and vegetation shadow, and the tradeoff between speed and accuracy when using different color enhancing methods. These attempts could provide useful reference for streamflow observation at other sites facing similar challenges. In the revised version of the manuscript, the authors have addressed most of the concerns raised by previous reviewers. I recommend acceptance after minor revisions.

(1) "acoustic doppler current profiler", the word doppler should be capitalized.

(2) In Introduction, the authors have focused on explaining image methods: PIV, PTV, STIV. Have deep learning techniques ever been used in hydrological monitoring? Previous studies on this topic should be discussed in this section.

(3) L61, lacking should be replaced by lack.

(4) 3.3.2 Comparison of discharge models. What is "discharge model"? It needs to be clarified because you have used "discharge classification model" through the paper.

---

## Author Response (AR2)

**Reviewer #4**

The manuscript presents a low-cost method with deep learning method to monitor discharge of mountain streams. Excellent performance was achieved with several preprocessing methods and coupled deep learning methods. However, there are still some issues that need further revision.

Thank you for reviewing our manuscript. Your constructive suggestions are crucial for enhancing its quality. We sincerely appreciate your time and effort, and we have carefully considered all of your suggestions. Below are our responses to your questions and our revisions to the manuscript.

1. The author coupled RF and SVM with CNN. However, training methods of these models are different, which means they cannot be trained together. Please provide details about how do you train coupled models.

Re: Thank you for your reminder. We acknowledge that while we did provide individual training details for the two coupled models (CNN+SVM and CNN+RF) in their respective sections (L314-316 for CNN+SVM and L325-326 for CNN+RF), our descriptions were overly ambiguous, which may have caused confusion among readers. Therefore, we have revised and clarified these sections to address their differences as follows:

(1) CNN+SVM (the training procedure is applied to SVM): The extracted image features, coded with a "one-vs-all" scheme, were used to train binary SVM classifiers. Specifically, one SVM classifier with a linear kernel function was trained for each discharge class to distinguish that class from the rest. The hinge loss function was employed to optimize the entire model by maximizing the margin between discharge classes.

(2) CNN+RF (the training procedure is applied to RF): We here used an RF comprising 350 decision trees and five decision leaves for discharge calculation. The coupling method of CNN+RF mirrors that of CNN+SVM, using the same pooling outputs of CNN as inputs for RF discharge classifier. RF is trained to assign optimal weights to each decision tree and leaf without a specific loss function.

2. Accuracy and F1 score are generally used for classification evaluation, while $R^2$ and RMSE are used for regression evaluation. In this study, the author conducted a classification task for discharge. It is unsuitable to use $R^2$ and RMSE for evaluating the results because the predicted discharge is discrete but the ground truth is continuous.

Re: Thank you for your feedback. Accuracy and F1 score are indeed commonly used in classification tasks. Although the discharge is treated as discrete in this study, we also consider the value difference between the simulated discharge and the flowmeter's discharge, especially if model simulations exhibit significant bias from the ground truth. This aspect cannot be adequately captured by accuracy and F1 score alone.

For instance, it was observed that the accuracy of $CNN+RF_{BZ}$ is 4.8% higher than that of $CNN_{CE}$, while the RMSE is also 0.05 m higher. This result suggests that while $CNN+RF_{BZ}$ has a higher likelihood of recognizing the true discharge, it also has a greater chance of identifying incorrect discharges with significant bias. Therefore, evaluating the model's performance solely based on accuracy and F1 score is incomplete. Incorporating regression metrics like $R^2$ and RMSE is appropriate to reflect the model's robustness in handling extreme outliers, which is equally crucial for discharge monitoring.

3. Lines 354-358, how did the author get the $-\bar{R} + 7.5\bar{G} - 6.5\bar{B}$ ? Is there any theory of this correlation? Please provide details about how to get the equation in the "Methods" section

Re: Thank you for your feedback regarding Section 3.1. Our aim in this section was to explore the relationship between image R/G/B characteristics and discharge values. Initially, we established the equation $a\bar{R} + b\bar{G} + c\bar{B}$, where coefficients $a$, $b$, and $c$ were to be determined. Through a systematic exploration of various combinations of $a$, $b$, and $c$, we identified that the characteristics derived from $-\bar{R} + 7.5\bar{G} - 6.5\bar{B}$ exhibit a high correlation with discharge values. This finding supports our assertion that discharge can be inferred directly from RGB matrices without the need for preliminary extraction of cross-sections and flow velocity, which forms the theoretical foundation of our study.

We acknowledge that this section serves as an introduction to subsequent sections. Our forthcoming work will leverage more sophisticated deep learning models to enhance the retrieval of discharge-related features under dynamic environmental conditions, ensuring robustness and stability.

In response to your suggestions for improved readability, we have incorporated additional details in Section 2.4, explaining how to get the equation.

4. The section "Correlation analysis" is not related to the theme of this study and not included in the flowchart. Why did the author build such a linear relationship in the study focused on deep

learning application?

Re: As previously explained, this section is essential as it serves as a preliminary step to demonstrate the feasibility of deriving flow discharge directly from RGB matrices without first extracting cross-sections and flow velocity. This foundational work establishes that discharge-related features are indeed embedded in images, thereby justifying and facilitating the subsequent use of deep learning models to extract flow discharge.

To aid in reader comprehension and avoid confusion, we have added the aim of this section to the manuscript.

5. Lines 460-463, is preprocessing time included in the comparison?

Re: The preprocessing time is not included in the comparison. The mentioned time refers to the duration taken by the three models to calculate discharge with preprocessed images as input. It is worth noting that preprocessing is significantly faster than discharge computation (approximately 10 times faster), which is why it was not considered in the overall timing.

To enhance readability and clarity, we have revised the description in the manuscript.

6. One of the application limitations of the study is that it could not be applied without labeled discharge from a gauge. It would be better to discuss more about the limitation and further improvement in the "Discussion" section.

Re: Thank you for your very constructive suggestion. We have added a further discussion on the limitations of our method and outlined future directions in Section 4, as follows:

"Moreover, site-specific field data is crucial for identifying the criteria for image categorization and model training, which restricts the broader applicability of our approach in ungauged basins, where such field data may not be readily available. Further research on integrating multiple data sources and surveying approaches is warranted for developing a more generalizable method."

**Reviewer #3**

(1) "acoustic doppler current profiler", the word doppler should be capitalized.

Re: Thank you. We have corrected it in the revised manuscript.

(2) In Introduction, the authors have focused on explaining image methods: PIV, PTV, STIV. Have deep learning techniques ever been used in hydrological monitoring? Previous studies on this topic should be discussed in this section.

Re: Thank you for your suggestion. In the Introduction, we have indeed focused on widely used image-based methods such as PIV, PTV, and STIV, which often lack applicability for mountain streams. Our study is the first attempt to retrieve flow discharge directly from RGB images without prior knowledge of river geometry and cross-sections using deep learning models, thereby addressing the challenges faced by these popular methods. Hence, we emphasized the comparison with these image-based methods.

We acknowledge that previous studies on deep learning techniques in hydrological monitoring are rare. However, we have discussed their promising application potential in Lines 83-90. Additionally, we identified a relevant study titled "RivQNet: Deep Learning Based River Discharge Estimation Using Close-Range Water Surface Imagery" (Ansari et al., 2023), which introduces the application of deep learning methods in river monitoring. This study demonstrates the representativeness and advantages of deep learning models, although the discharge estimation still relies on cross-section information and derives surface velocity using CNN.

We have incorporated a brief discussion of this literature in the Introduction to highlight the advantages of deep learning models.

(3) L61, lacking should be replaced by lack.

Re: Thank you. We have corrected it in the revised manuscript.

(4) 3.3.2 Comparison of discharge models. What is "discharge model"? It needs to be clarified because you have used "discharge classification model" through the paper.

Re: Thank you. We have standardized its terminology throughout the text to "discharge classification model(s)".

**Reference:**

Ansari, S., Rennie, C., Jamieson, E., Seidou, O., and Clark, S.: RivQNet: Deep Learning Based River Discharge Estimation Using Close‑Range Water Surface Imagery, Water Resources Research, 59, 10.1029/2021WR031841, 2023.

Ansari, S., Rennie, C., Jamieson, E., Seidou, O., and Clark, S.: RivQNet: Deep Learning Based River Discharge Estimation Using Close‑Range Water Surface Imagery, Water Resources Research, 59, 10.1029/2021WR031841, 2023.

**Reviewer #2**

1. Lines 98-100. The first hypothesis "the features of mountain streams (e.g., coverage of water surface, flow direction, flow velocity) embedded in RGB images can be recognized..." is better to be responded or discussed in the discussion or conclusion section.

Re: Thanks for your constructive suggestions. We have addressed the two hypotheses in the "Conclusion" section to better summarize our work and facilitate reader understanding.

2. Section 2.2. I suggest a hydrograph for the period July 20th to September 27th 2022 with high quality image data marked.

Re: Thank you for your constructive suggestions. To improve the presentation, we have added a hydrograph demonstrating the in-situ discharge with excellent quality to Figure 1, as follows:

[Figure]

Figure 1. Camera setup. The camera is set on the left bank of the stream, about 3 m above the water surface, and 8 m upstream of a gauging weir. The top right panel demonstrates the changes in the flowmeter's discharge during the measurement period.

3. Line 333. "Accuracy" should be "Classification accuracy" and in other sections.

Re: Thank you very much. We have standardized the terminology to "Classification accuracy" when referring to the performance of our discharge classification models. This includes updating the labels

in Figures 6 and 8.

4. The discussion section is about the advantage, limitations, and the role of key procedures of the new method. It is better to reorganize this section to make it clearer.

Re: Thank you for your constructive suggestions. We have reorganized the Discussion section to make it clearer by presenting the advantages, limitations, and potential directions for future improvement in that order. Additionally, we have added some content highlighting the primary limitation and suggesting possible improvements.

5. Lines 539-541. The general performance of the method evaluated by R2 and RMSE is needed.

Re: Thanks for your reminder. We have included the comparison of $R^2$ and RMSE as follows,

"In this case, the classification accuracy, F1 score, and R2 of CNN+SVM and CNN+RF were 9.1%~14.4%, 0.084~0.115, and 0.006~0.010 higher, respectively, while RMSE was 0.31~0.51 m lower compared to CNN."

Additionally, we have supplemented the demonstration of these metrics in Sections 3.3.2 and 3.3.3 to facilitate better comparison and provide a clearer highlight of the best models and color-enhancing methods.

**Reviewer #1**

The authors have not yet answered the question of how the proposed method can be applied to practical flow measurements. Moreover, 37 traffic samples are too small, so the sample data in this paper are not representative.

During floods, the light is dark and storms often occur. Therefore, the traffic monitoring method established only from the color classification may be feasible in a certain condition, but it is difficult to apply in more scenarios. The paper lacks sufficient traffic monitoring data to verify the rationality of the results under different environmental conditions, so it is recommended to be rejected.

Re: Thank you for your comment. Discharge monitoring at rocky upstream mountain streams has been a difficult task for a long time due to the complex topography. We agree that the samples presented in this study did not cover all environmental conditions, which affects the applicability and transferability of the models. We have thoroughly discussed about these limitations in the manuscript. However, we believe that our study represents a new direction for applying deep learning techniques in acquiring high-frequency discharge data through image analysis. Although the method was only tested at one single site, it provides a different idea that could serve as an alternative apparatus, or integrated into traditional monitoring approaches to improve data quality. The paper also tackled a few important issues in streamflow image processing, including the treatment of images affected by the disturbances of water reflection and vegetation shadow, and the tradeoff between speed and accuracy when using different color enhancing methods. These attempts could provide useful reference for streamflow observation at other sites facing similar challenges.